# Controlling Coverage of Uncertainty Sets for Batch Evaluation via Vanilla Conformal Prediction

## Abstract

Conformal prediction (CP) provides provable coverage guarantees over uncertainty sets for any given black-box predictive model. The standard split CP guarantees that for a single test input, the uncertainty set contains the true output with a user-specified probability $1-\alpha$ (say 90%). However, in many real-world applications, practitioners evaluate the predictive model on a batch of test inputs after calibration on a fixed set. The marginal coverage guarantee of split CP does not say anything directly about the realized false-coverage proportion (FCP) across a batch of inputs. This paper develops *a simple and effective* approach referred to as *Probably Approximately Correct FCP (PAC-FCP)*. PAC-FCP leverages the key insight that FCP over a batch of test inputs from split CP follows a Beta-Binomial distribution and inverts the Beta–Binomial tail to find the minimum level to produce a guarantee around FCP using vanilla CP methods. We provide theoretical analysis for the validity and effectiveness of PAC-FCP *using* prior theoretical results. Our experimental results on 17 OpenML benchmarks for regression and ImageNet data for classification, demonstrate that PAC-FCP achieves the specified FCP rate with smaller prediction sets/intervals.

## 1 INTRODUCTION

Safe deployment of machine learning (ML) models in high-stakes applications such as medical diagnosis requires theoretically sound uncertainty quantification (UQ). Conformal prediction (CP) Vovk et al. (2005); Lei & Wasserman (2014); Angelopoulos & Bates (2022) is a promising framework for uncertainty quantification that wraps around any black-box predictive model and returns a prediction set/interval for classification/regression tasks with a finite-sample, distribution-free, and model-agnostic coverage guarantee. Split CP is the most commonly employed instantiation for ML applications. Given an ML-based predictive model, a calibration set of $n$ input-output pairs, and a user-specified target miscoverage level $\alpha$ (say 10%), split CP produces prediction sets whose marginal coverage is guaranteed:

$$\Pr\left(Y_{n+1} \in C_\alpha(X_{n+1})\right) \geq 1 - \alpha, \tag{1}$$

where $(X_{n+1}, Y_{n+1})$ is a new test input, $C_\alpha(X_{n+1})$ is the prediction set, and $Y_{n+1}$ is the true output. This probability is taken with respect to the joint randomness of the calibration examples and the future test input (aka *exchangeable sampling*): on average over all such draws, the prediction set contains the true output with probability at least $1 - \alpha$.

**Motivation.** Unfortunately, this marginal guarantee (applicable for a single test input) is not useful in real-world applications of ML models where practitioners typically evaluate a batch of $t$ testing inputs $(X_{n+1}, Y_{n+1}), \ldots, (X_{n+t}, Y_{n+t})$ (*batch evaluation*), e.g., multiple patients in health diagnosis. Practitioners often (and understandably) interpret the CP's marginal coverage guarantee as implying control over the fraction of errors in such a batch. Suppose $\alpha=0.1$ and $t=100$, it is tempting — but incorrect — to read this as "at most ten of the next 100 predictions will be wrong." Specifically, the marginal guarantee does *not* directly translate into a statement about the realized fraction of errors within a batch: re-using the same calibration set for all $t$ test inputs *breaks the exchangeability assumption* ($t$ coverage events are *not* independent) and does not specify how often coverage fails. Indeed, even as $t \to \infty$, the practitioner

may observe arbitrarily long streaks of errors or perfect coverage, because the latent coverage probability is a random quantity informed by the single draw of calibration examples. Thus, in practical ML model deployments, marginal coverage alone does not answer the key question practitioners face: "How many of these $t$ predictions should I expect to be wrong?"

Practitioners require a bound on the *false coverage proportion* (FCP), i.e. the realised fraction $\widehat{\text{FCP}} = K/t$ with $K$ mis-covered inputs among an evaluation batch of $t$ test inputs. $\widehat{\text{FCP}}$ is the only directly observable quantity after deployment, and it is the metric users rely on when assessing the reliability of ML models. Controlling $\widehat{\text{FCP}}$ is the only operational guarantee visible to the user after ML model deployment, but it is precisely what the marginal guarantee in CP leaves unspecified. The problem of controlling FCP for batch evaluation hasn't received attention from the CP community until recently Gazin et al. (2024); Blain et al. (2025). *A very recent work Blain et al. (2025) proposed the CoJER method as an improvement over Gazin et al. (2024) to provide a uniform-in-$\alpha$ guarantee that practitioners can use to control FCP. However, the stronger uniform-in-$\alpha$ guarantee comes with* conservatism and algorithmic complexity resulting in large prediction sets *which are not useful in real-world applications Babbar et al. (2022); Straitouri et al. (2023).*

This paper *develops a simple and effective* approach referred to as *Probably Approximately Correct FCP (PAC-FCP)* to address the challenge of controlling coverage for batch evaluation. PAC-FCP is a simple approach that provides a PAC-style guarantee $\Pr(\widehat{\text{FCP}} \leq \alpha) \geq 1 - \delta$ for any user-chosen $(\alpha, \delta)$ and batch size $t$, where $\delta$ is the confidence parameter. The key insight behind PAC-FCP is the observation that conditioned on the calibration set, the coverage indicators of split CP for testing inputs follow a Beta–Binomial predictive distribution. By inverting this distribution, we can translate an internal guarantee of split CP into an $(\alpha, \delta)$ high-probability guarantee on batch-level FCP:

$$\Pr\left(\frac{K}{t} \leq \alpha\right) \geq 1 - \delta, \tag{2}$$

where $K$ is the number of mis-covered inputs in the batch of size $t$. Importantly, PAC-FCP uses nothing beyond vanilla split CP machinery and the i.i.d. assumption for testing inputs. It requires no uniform-in-$\alpha$ control, no sequential corrections, and no additional re-weighting steps, making it straightforward to integrate into existing vanilla CP pipelines. Our experiments on 17 OpenML datasets for regression and ImageNet data for classification show that PAC-FCP achieves the user-specified FCP and produces significantly smaller prediction intervals/sets compared to CoJER.

**Contributions.** The key contribution is the development and evaluation of the PA-FCP approach to control false-coverage proportion over a batch of inputs. Specific contributions include:

- Defining a practically motivated and ML model deployment-relevant relaxed guarantee for *evaluation-batch FCP control* problem in CP.

- Development of the PAC-FCP approach, a lightweight calibration procedure that reuses standard split CP machinery and requires no additional modeling or sequential corrections, while producing smaller prediction sets/intervals for any given FCP rate.

- Theoretical analysis of PAC-FCP for validity and effectiveness to control FCP *using prior theoretical result that distribution of coverage has an analytical form Vovk (2012); Angelopoulos & Bates (2022); Corollary A.3 in Gazin et al. (2024); and Theorem 1 in Marques F. (2025). This result appears to be a special case* [1] *of Theorem B.1 in the Appendix of Gazin et al. (2024) with the* Beta template.

- Empirical evaluation of PAC-FCP and comparison with baselines on 17 OpenML benchmarks for regression and ImageNet data for classification.

---

[1] *Thanks to a knowledgeable reviewer who pointed out this connection to us.*

## 2 RELATED WORK

Conformal prediction (CP) provides distribution-free, finite-sample guarantees under exchangeability and has seen broad use in classification Romano et al. (2020); Sadinle et al. (2019) and regression Romano et al. (2019); Papadopoulos et al. (2002) settings. Motivated by the need for small prediction sets/intervals in real-world applications, much of the recent work in CP is focused on improving predictive efficiency via improved nonconformity scores Angelopoulos et al. (2021) and calibration methods Ghosh et al. (2023). There is also some work on CP-aware training, where models are trained to optimize conformal objectives so that post-hoc CP yields smaller prediction sets Stutz et al. (2022); Einbinder et al. (2022).

Another line of work is to strengthen guarantees beyond marginal coverage, such as group-conditional Ding et al. (2024); Gibbs et al. (2023) or coverage under distribution shifts Tibshirani et al. (2019) and different forms of approximate conditional coverage Guan (2023); Jung et al. (2022); Gibbs et al. (2023). A complementary research direction targets *risk control* using CP ingredients, to control different types of risks for different problem setups Angelopoulos et al. (2025); Bates et al. (2021); Angelopoulos et al. (2022).

Except for the very recent work from Blain et al. (2025) and Gazin et al. (2024) *that provides the stronger uniform-in-$\alpha$ guarantee*, prior CP literature didn't address the practical need for CP methods that provide coverage guarantees for a batch of testing inputs by producing small prediction sets. Our work precisely addresses this critical gap.

## 3 PROBLEM SETUP

We first introduce notations and background on split conformal prediction. Next, we formally define the problem setup and discuss the closest prior work.

### 3.1 Notations and Background

**Notations.** Suppose $X \in \mathcal{X}$ is an input from $\mathcal{X}$, and $Y \in \mathcal{Y}$ is the ground-truth output, where $\mathcal{Y}$ is continuous for regression tasks and is discrete ($K$ candidate class labels) for classification tasks. Let $(X, Y)$ be a data sample drawn from an underlying distribution $\mathcal{P}$ defined on $\mathcal{X} \times \mathcal{Y}$. We assume the availability of a trained *predictive model $f$* (classifier or regressor) a *calibration set* of $n$ examples $D_{\text{cal}} = \{(X_i, Y_i)\}_{i=1}^n$. The given predictive model $f$ needs to be calibrated to produce prediction sets/intervals to quantify its uncertainty on testing inputs. We will evaluate the resulting prediction sets on an *evaluation batch* of size $t$,

$$D_{\text{test}} = \{(X_{n+j}, Y_{n+j})\}_{j=1}^t.$$

We define the *miscoverage indicator* $Z_j = 1\{Y_{n+j} \notin C(X_{n+j})\}$, where $C(X_{n+j})$ is the prediction set/interval of test input $X_{n+j}$. The total number of mis-coverages is $K = \sum_{j=1}^t Z_j$ and the realized FCP is defined as

$$\widehat{\text{FCP}} = \frac{K}{t}.$$

**Split CP.** For the sake of exposition and ease of understanding, we assume regression tasks. Split CP Angelopoulos & Bates (2022) allows us to compute a prediction interval for any given predictor through a conformalization step using calibration data. First, we choose a conformity score function $S(X, Y)$ to measure how the model's output ($f(X)$) for input $X$ conforms to the true output $Y$. For regression tasks, the absolute residual $S(X, Y) = |f(X) - Y|$ is a typical conformity score. Second, we compute conformity scores $S_i = S(X_i, Y_i)$ for all calibration pairs in $D_{\text{cal}}$. Next, we compute $k = 1 + \lfloor (n+1)(1-\alpha) \rfloor$ and set $q_{1-\alpha}$ to the k$^{\text{th}}$ smallest value in $\{S_i\}_{i=1}^n$ as the calibration threshold to achieve the target coverage $1 - \alpha$. For a new testing input $X$, split CP computes the prediction interval as follows:

$$C_\alpha(X) = \{Y : S(X, Y) \leq q_{1-\alpha}\}.$$

Under the assumption of *exchangeability* (the order in which the data samples appear doesn't matter for their joint distribution), this guarantees the marginal coverage property Lei & Wasserman (2014)

$$\Pr\left(Y_{n+1} \in C_\alpha(X_{n+1})\right) \geq 1 - \alpha.$$

This probability is over the joint sampling of calibration examples and future testing inputs. Once the calibration set $D_{\text{cal}}$ is fixed and we reuse it, the guarantee does not automatically translate to the common setting in practice, where we evaluate a batch of inputs.

## 3.2 Problem Setup and Closest Prior Work

We consider the setup where the realized performance on the entire *evaluation batch* $D_{\text{test}}$ is required and *not* just one testing input. This is common in almost all real-world deployment of ML models (e.g., multiple patients in health diagnosis). Therefore, we focus on the statistic $\widehat{\text{FCP}} = K/t$, the proportion of mis-covered inputs in $D_{\text{test}}$, and seek a high-probability upper bound on this realized quantity.

Concretely, given a user-specified $(\alpha, \delta)$ and batch size $t$, we want a calibration procedure that ensures

$$\Pr\left(\frac{K}{t} \leq \alpha\right) \geq 1 - \delta,$$

where the probability is with respect to the sampling of all data (training, calibration, and testing), but inside the probability statement is about the actual realized FCP. CoJER provides a guarantee of the following form:

$$\Pr\left(\forall \alpha \in [0,1], \ \text{FCP}(C_{\alpha^*}) \leq \alpha\right) \geq 1 - \delta \tag{3}$$

*for some value $\alpha^*$ determined as a function of the problem parameters, which defines conformal sets at level $\alpha^*$ such that they satisfy the uniform-in-$\alpha$ guarantee in Equation (3). CoJER Blain et al. (2025) uses conformal p-values with joint error rate control to achieve this guarantee. Because the uniform-in-$\alpha$ guarantee in Equation (3) protects all $\alpha$ simultaneously, it produces large prediction intervals, which are undesirable in practical applications Babbar et al. (2022); Straitouri et al. (2023).*

In our setting, the practitioner fixes $(\alpha, \delta, t)$ *before* evaluation (just as they fix $\alpha$ in vanilla split CP) and seeks a high-probability bound on the *realized* FCP on a single evaluation batch. This is weaker than CoJER's uniform, post-hoc guarantee, but is often exactly what is required in practice. Our goal is to develop an algorithm that produces smaller prediction intervals by avoiding the price paid for a uniform guarantee over $\alpha$ as in CoJER.

# 4 PAC-FCP ALGORITHM

In this section, we first provide a high-level overview of our proposed PAC-FCP approach. Next, we describe the key intuition behind PAC-FCP. Finally, we explain the key algorithmic steps of PAC-FCP.

**Overview of PAC-FCP.** PAC-FCP exploits the calibration-conditional behavior of the split conformal prediction approach to obtain a high-probability bound on the realized false coverage proportion (FCP) over an *evaluation batch* of size $t$. Given target parameters $(\alpha, \delta)$ and a calibration set $\mathcal{D}_{\text{cal}}$, PAC-FCP finds an $\alpha' \in (0,1)$ such that the split CP algorithm run at $\alpha := \alpha'$ satisfies the Eq. 2, the guarantee required in the problem setup. The key element to achieve this goal is the Beta-Binomial Quantile function, which has a closed-form (see Line 5 in Algorithm 1).

## 4.1 Main Intuition and Sketch of PAC-FCP

The key idea builds on the exchangeability principle in conformal prediction: the $n+1$ score ranks are uniformly distributed, and this uniformity of the new test input's rank underpins the marginal coverage guarantee via the typical $n+1$ argument Angelopoulos & Bates (2022). When we *condition* on the first $n$ examples, the calibration set $\mathcal{D}_{\text{cal},0} = \{(X_i, Y_i)\}_{i=1}^n$ *sampled from the data distribution* fixes the prediction set map $C_\alpha(\cdot)$. However, across different draws of $\mathcal{D}_{\text{cal}}$, the induced prediction set map changes and so does the conditional coverage probability, leading to different values of $\Pr\left(Y \in C_\alpha(X) \mid \mathcal{D}_{\text{cal}} = \mathcal{D}_{\text{cal},0}\right)$. This

---

**Algorithm 1:** PAC-FCPApproach

---

**Input:** batch size $t \in \mathbb{N}$,
target $(\alpha, \delta)$ with $\alpha \in (0,1)$, $\delta \in (0,1)$; calibration set $\mathcal{D}_{\text{cal}} = \{(X_i, Y_i)\}_{i=1}^n$; predictive model $f$ and nonconformity score $S(X, Y)$; numerical tolerance $\varepsilon > 0$.
**Output:** Internal level $\alpha'$, quantile $q_{1-\alpha'}$, and prediction-set map $C_{\alpha'}(\cdot)$ satisfying
      $\Pr\big(K/t \leq \alpha\big) \geq 1 - \delta$.

**1 Compute conformity score of calibration examples.**
**2 for** $i \leftarrow 1$ **to** $n$ **do**
**3**     $S_i \leftarrow S(X_i, Y_i)$
**4** Sort $\{S_i\}_{i=1}^n$ in non-decreasing order: $S_{(1)} \leq \cdots \leq S_{(n)}$.
**5 Define the Beta–Binomial Quantile (BBQ) function.**
**6** For any $\alpha' \in (0,1)$, let $\ell(\alpha') \leftarrow \lfloor \alpha'(n+1) \rfloor$.
**7** Define $F_{\text{BBQ}}(\alpha')$ as

$$\sum_{k=0}^{\lfloor \alpha t \rfloor} \binom{t}{k} \frac{B\big(k + \ell(\alpha'),\ t - k + n + 1 - \ell(\alpha')\big)}{B\big(\ell(\alpha'),\ n + 1 - \ell(\alpha')\big)},$$

    the CDF of Beta–Binomial$(t, \ell(.), n + 1 - \ell(.))$ at $t - \lfloor \alpha t \rfloor$.

**8 Find the largest feasible $\alpha'$ to achieve the target $(\alpha, \delta)$.**
**9** Find the largest feasible $\alpha'$ such that $F_{\text{BBQ}}$ is lower than $\delta$.

**10 Compute conformal quantile and prediction sets.**
**11** $k^\star \leftarrow 1 + \lfloor (n+1)(1 - \alpha') \rfloor$
**12** Quantile $q_{1-\alpha'} \leftarrow S_{(k^\star)}$
**13** Define prediction set mapper as

$$C_{\alpha'}(X) = \{Y : S(X, Y) \leq q_{1-\alpha'}\}.$$

**14 return** $\alpha'$, $q_{1-\alpha'}$, and $C_{\alpha'}(\cdot)$.

---

intuition can be formalized to show that the conditional coverage probability follows a Beta distribution. Formally, we write

$$\theta = \Pr\big(Y_{n+1} \in C_\alpha(X_{n+1}) \,\big|\, \mathcal{D}_{\text{cal}}\big)$$

is Beta–distributed with known parameters (as shown in Theorem 1). Figure 1 is a conceptual example that illustrates how and why this randomness exists.

Given $\mathcal{D}_{\text{cal}} = \mathcal{D}_{\text{cal},0}$, the conformal prediction set map $C_\alpha(\cdot)$ is fixed, and has a fixed coverage probability $\theta = \theta_0$. Therefore, the indicators for testing inputs

$$Z_j = \mathbf{1}\{Y_{n+j} \notin C_\alpha(X_{n+j})\}, \qquad j = 1, \ldots, t,$$

are conditionally i.i.d. Bernoulli$(1 - \theta_0)$. By integrating over $\theta$ space to remove the conditioning on $\mathcal{D}_{\text{cal}}$, the coverage or mis-coverage count $K = \sum_{j=1}^t Z_j$ follows a Beta–Binomial distribution with known parameters. Consequently,

$$\Pr\big(\widehat{\text{FCP}} \leq \alpha\big) = \Pr\big(K \leq \lfloor \alpha t \rfloor\big)$$

is available in closed form via the Beta–Binomial CDF. See Figure 2 for a conceptual illustration.

PAC-FCP then searches over the internal level $\alpha'$ of split CP to find the smallest value for which the Beta–Binomial Quantile (BBQ) satisfies the target guarantee,

$$\Pr\big(K \leq \lfloor \alpha t \rfloor\big) \geq 1 - \delta,$$

to deliver the desired PAC-style control of *FCP*.

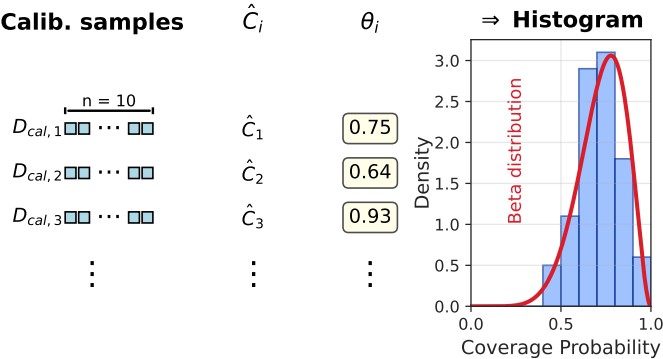

Figure 1: **Illustration of calibration data-conditioned coverage probability distribution**. Given different samples for the calibration set, we obtain different prediction set maps $\hat{C}_i$ with different coverage probabilities for the $(n+1)$-th testing input. This randomness creates a Beta distribution for the conditional probability $\theta = \Pr(y \in \hat{C}|\mathcal{D}_{\text{cal},})$. This plot shows results for $\hat{C}_i$ created using split conformal prediction with $\alpha = 0.3$ and $n = 10$.

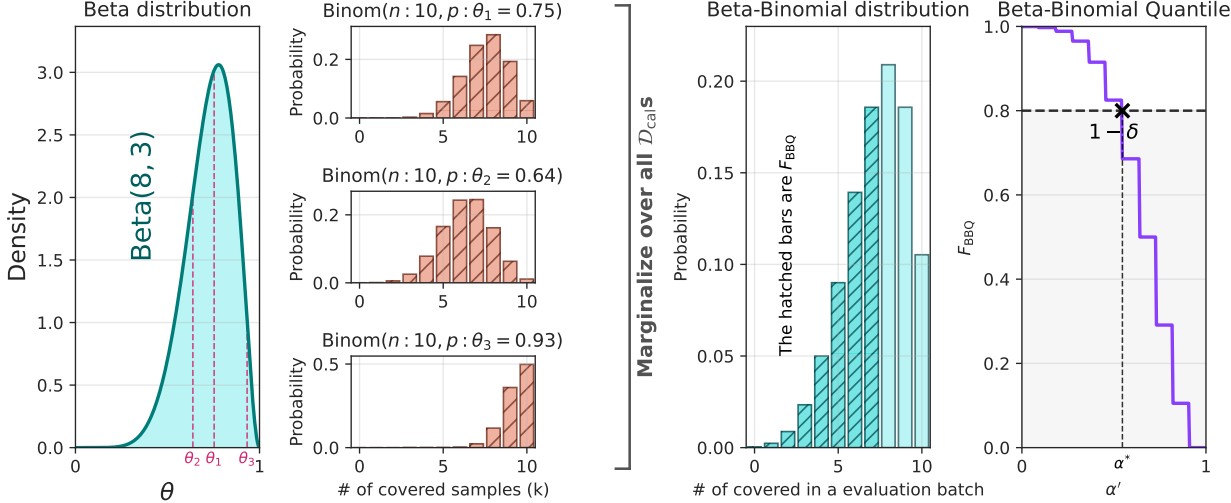

Figure 2: **Illustration of Beta-Binomial Quantile (BBQ) function and the intuition for PAC-FCP**. Each calibration sample leads to a different value of $\theta$. When sampling $t = 10$ test inputs, the number of observed covered testing inputs follows a $\text{Binom}(t, \theta)$ distribution. Marginalizing over all possible calibration data samples, the number of covered test inputs in conformal prediction follows a Beta-Binomial distribution. The plot uses the same split-CP configuration as in Figure 1. The rightmost column shows the *Beta–Binomial quantile (BBQ)* function defined in Algorithm 1; the curve reports the fraction of calibration datasets for which the evaluation batch fails (miscoverage proportion $> \alpha$) when SCP is run at level $\alpha'$ (instead of $\alpha$). PAC-FCP selects the largest level $\alpha^\star$ where this failure rate falls below $\delta$, thereby guaranteeing Equation (2).

## 4.2 Key Algorithmic Steps of PAC-FCP

Algorithm 1 shows the pseudocode of the PAC-FCP approach. It follows the typical workflow of split CP with one extra knob: in addition to a miscoverage target $\alpha$, we also accept a confidence level $\delta$. Below, we explain the key algorithmic steps of PAC-FCP.

**Compute conformity scores.** We begin the workflow exactly as in Split CP. Using the given predictive model $f$ and conformity scoring function $S(X, Y)$, we compute the conformity scores $S_i = S(X_i, Y_i)$ over the provided calibration examples $\mathcal{D}_{\text{cal}} = \{(X_i, Y_i)\}_{i=1}^n$ and sort them in non-increasing order (Lines 1-3).

**Define Beta-Binomial Quantile Function** $F_{\text{BBQ}}$. Line 5 of the pseudocode defines $F_{\text{BB}}(\alpha')$, the Beta–Binomial tail probability of observing at most $\lfloor \alpha t \rfloor$ mis-coverages $K$ in a batch with $t$ testing inputs if we were to run split CP at internal level $\alpha'$ (see Section 5 for more details). Intuitively, $F_{\text{BBQ}}(\cdots)$ asks: "With this $\alpha'$, how likely am I to meet the target batch-level coverage rate of $\alpha$?"

**Choosing $\alpha'$.** PAC-FCP performs grid search over the interval $[0, 1)$ with step $\varepsilon = 0.005$ and selects the *largest* $\alpha'$ that satisfies $F_{\text{BB}}(\alpha') \geq 1 - \delta$ in Line 5. Any one-dimensional root-finding method for solving can be employed for this purpose. We adopt a coarse grid because verifying $F_{\text{BB}}(\alpha') \geq 1 - \delta$ is efficient. As shown in Figure 2, the space is discrete because the parameters of the Beta Binomial distribution in Theorem 1 take discrete values.

**Computing prediction sets.** With the chosen $\alpha'$ in hand, we retrieve the split CP threshold, $q_{1-\alpha'}$ as the $\lceil (n+1)(1-\alpha')/n \rceil$ quantile of the conformity scores of calibration examples, and form the map for prediction set $C_{\alpha'}(X)$. For absolute-residual scores, this is the symmetric interval $[f(X) - q_{1-\alpha'}, \ f(X) + q_{1-\alpha'}]$.

## 5 THEORETICAL ANALYSIS

In this section, we analyze the theoretical properties of PAC-FCP as implemented in Algorithm 1.

**Validity of Coverage from PAC-FCP.** The marginal coverage guarantee of split CP follows from the exchangeability of the $n{+}1$ rank statistics: with the typical $n{+}1$ correction, the (randomized) order statistic threshold ensures that the new testing input's conformity score falls below the calibration quantile with the prescribed probability. Adapting this rank view to the batch evaluation setting yields the following Beta-Binomial distribution for coverage *that is known from prior work including Vovk (2012); Angelopoulos & Bates (2022);Corollary A.3 in Gazin et al. (2024);Theorem 1 in Marques F. (2025). We restate this theoretical result in the context of our specific problem setting below.*

**Theorem 1** (Batch miscoverage is Beta–Binomial). *Let $C_\alpha$ be a conformal method with level $\alpha$, on i.i.d data, with a continuous conformity score (defined as $S$) and randomized ties. Let $Z_j = \mathbf{1}\{Y_{n+j} \notin C_\alpha(X_{n+j})\}$ for $j = 1, \ldots, t$, and $K = \sum_{j=1}^{t} Z_j$. And define $\ell(\alpha) := \lfloor \alpha(n+1) \rfloor$.*

*The predictive distribution of $K$ over an evaluation batch of size $t$ is Beta-Binomial.*

$$\Pr(K = k) \ = \ \binom{t}{k} \frac{B(k+\ell, \ t-k+n+1-\ell)}{B(\ell, \ n+1-\ell)},$$

*where $B(\cdot, \cdot)$ is the Beta function and $k \in \{0, \ldots, t\}$.*

This result is the key ingredient to provide guarantees for PAC-FCP: in a batch evaluation setting with $t$ testing inputs, the error count $K$ has a known, distribution-free predictive law determined by $(\alpha', n, t)$, allowing direct computation of the tail probabilities $\Pr(K \leq \lfloor \alpha t \rfloor)$.

Theorem 1 induces the function $F_{\text{BB}}(\alpha') := \Pr(K \leq \lfloor \alpha t \rfloor)$, leading to the PAC guarantee for Algorithm 1.

**Lemma 2** (PAC-style FCP control via tail inversion). *Fix $\alpha \in [0, 1]$, $\delta \in (0, 1)$, and batch size of testing inputs $t$. Let $\ell(\alpha') := \lfloor \alpha'(n+1) \rfloor$ for some internal split CP level $\alpha'$. If*

$$\sum_{k=0}^{\lfloor \alpha t \rfloor} \binom{t}{k} \frac{B(k+\ell, \ t-k+n+1-\ell)}{B(\ell, \ n+1-\ell)} \ \geq \ 1 - \delta, \tag{$\star$}$$

*Then, split CP run at level $\alpha'$ with a score that does not produce ties (e.g., via random tie-breaking) guarantees*

$$\Pr\left(\frac{K}{t} \leq \alpha\right) \ \geq \ 1 - \delta.$$

Thus, Algorithm 1 (PAC-FCP) provides the coverage validity guarantee in Eq. 2.

*This Lemma appears to be a special instantiation of Theorem B.1 in the Appendix of Gazin et al. (2024) (a more general Theorem) with the Beta template[2]. To the best of our knowledge, there is also no practical implementation of the variant with Beta template version of the approach from Gazin et al. (2024) with the uniform-in-$\alpha$ guarantee[3] to perform experiments.*

**Remark.** As $t \to \infty$, the $\frac{K}{t}$ goes toward the expectation value, which is the False Coverage Rate, FCR=$\mathbb{E}\{FCP\}$ Blain et al. (2025). Using this approach can also provide a guarantee, because in this scenario $\frac{K}{t} \to \theta$, so the method effectively reduces to using the Beta distribution to obtain an $(\alpha, \delta)$-guarantee around the Beta distribution.

It is worth mentioning that any SCP-like solution (i.e., prediction sets formed from order statistics of the calibration scores) that satisfies Equation (2) must use at least the same quantile as PAC-FCP, which leads to larger sets in both classification and regression. A more rigorous version of this statement appears in the Appendix along with all proofs.

# 6 EXPERIMENTS AND RESULTS

This section describes our experiments and results.

## 6.1 Experimental Setup

**Datasets.** For the regression task, we follow Blain et al. (2025), and employ the **17** regression tasks from the OpenML benchmark data Grinsztajn et al. (2022). For every $(\alpha, \delta)$ we evaluate, each dataset is randomly shuffled and split into *train*, *calibration*, and *test* subsets $n_{\text{split}}$ times. The exact value of $n_{\text{split}}$ is noted with each experiment.

For classification, following the CP literature Angelopoulos et al. (2021), we employ the Imagenet-Val Russakovsky et al. (2015), and a pretrained deep model. We split the validation data into *calibration* and *test* splits. We do the experiment $n_{\text{split}}$ times.

**ML models.** Similar to Blain et al. (2025), on every training split we fit five standard regressors for regression tasks: 1) multilayer perceptron (MLP) Hinton (1989), 2) support-vector regression (SVR) Platt (1999), 3) $k$-nearest neighbours (KNN) Cover & Hart (1967), 4) random forest (RF) Breiman (2001), and 5) Lasso regression Tibshirani (1996). For classification task, we adopt a pretrained ResNet He et al. (2015) model from PyTorch's torchvision implementation.

**Methods.** We benchmark three methods:

(i) **PAC-FCP** – the proposed procedure.

(ii) **SCP** – vanilla split conformal prediction Angelopoulos & Bates (2022), which doesn't come with evaluation-batch guarantee;

(iii) **CoJER** – method with a different guarantee shown in Equation (3) Blain et al. (2025).

(iv) ***Gazin et al.** – Gazin et al. (2024) is the prior work to Blain et al. (2025), that achieves a similar uniform-in-$\alpha$ guarantee Equation (3). It is generally more conservative than CoJER Blain et al. (2025) as demonstrated in our experiments as well.*

All methods employ the same calibration set and testing set for evaluation purposes in each run.

**Evaluation methodology.** For every dataset, we generate $n_{\text{split}}$ independently sampled *train / calibration / test* splits. Within each split, for regression only, we (i) train every model on the training set, (ii) calibrate the resulting predictor on the calibration set using each method, and (iii) evaluate the calibrated predictor on the test set. We employ $n_{\text{split}}$=40 for the regression results in Table 1, and $n_{\text{split}}$=10 for Figure 4. For

---

[2]*Thanks to a knowledgeable reviewer who pointed out this connection to us.*
[3]*https://github.com/ulyssegazin/TransductiveAdaptive_CP*

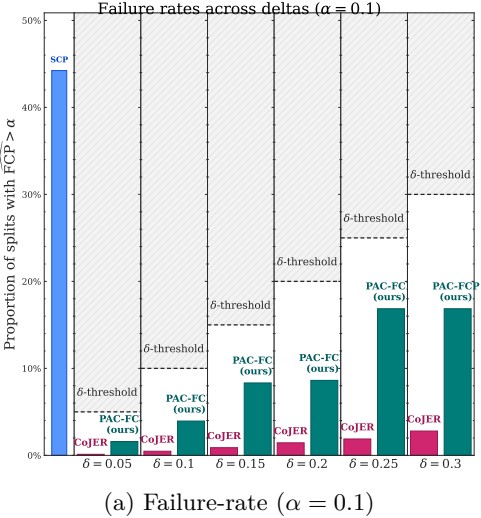
(a) Failure-rate ($\alpha = 0.1$)

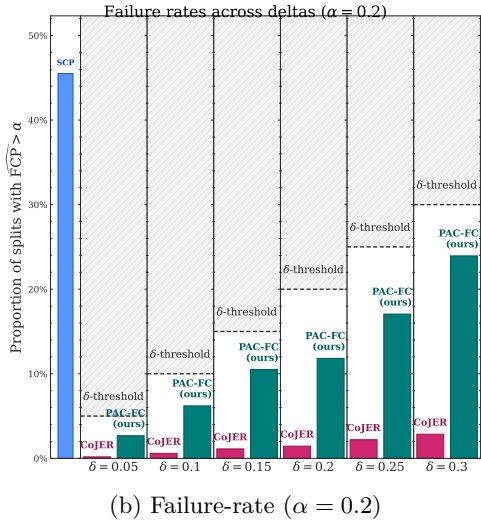
(b) Failure-rate ($\alpha = 0.2$)

Figure 3: **Failure rates across $\delta$ for two miscoverage levels.** Panels (a) and (b) correspond to $\alpha = 0.1$ and $\alpha = 0.2$, respectively. Within each subfigure, the left mini-panel shows the vanilla SCP baseline (invariant to $\delta$), and the rest of the mini-panels to the right compare PAC-FCP and CoJER in a range of $\delta$. Bars give the proportion of evaluation splits with $\widehat{\text{FCP}} > \alpha$; the dashed horizontal line marks the target $\delta$ from equation 2, with the hatched region above indicating the inadmissible zone for valid methods. Values are averaged within each run over 17 OpenML datasets $\times$ 5 models $\times$ $n_{\text{split}}$=40 splits, and aggregated across four independent runs.

classification, we set $n_{\text{split}} = 3000$. The classification results in Figure 4 and Table 1 are the average of five experiments with $n_{\text{split}}$ splits.

**Evaluation metrics.** For each randomized experiment, we record (i) whether the realized FCP *fails* the condition in Equation (2) (i.e. $\widehat{\text{FCP}} \geq \alpha$), and (ii) the average prediction interval length. Because raw interval lengths depend on the output scale of the dataset and the model, we normalize interval size by the interval length of vanilla SCP, yielding scale-free ratios that are comparable across tasks. The expected value of metric (i) should not exceed $\delta$; among the methods that meet this bound, values *closer to $\delta$* indicate tighter calibration, whereas markedly smaller values signal conservatism (over-coverage in the calibration space). For metric (ii), lower normalized interval lengths are always better, reflecting narrower prediction intervals at equal validity.

## 6.2 Results

**Validity of FCP control.** Our first experiment measures the validity of the tail condition from Equation equation $\star$ in PAC-FCP. Fixing $(\alpha, \delta)$, we repeat the following for every dataset–model pair: shuffle the data, split 2/3–2/9–1/9 into train/calibration/test, and evaluate on $n_{\text{split}} = 40$ random splits (yielding $17 \times 5 \times 40$ evaluation batches).

Table 1 summarizes the failure rates across $\delta \in \{0.05, 0.1, 0.2, 0.3, 0.4\}$ for $\alpha \in \{0.1, 0.2\}$. The *Failed Proportions* column of SCP lacks any batch-level guarantee: miscovers in roughly 44% (for regression) of splits in each $\alpha$. Using Theorem 1, the theoretical SCP failure probability at level $\alpha$ equals $1 - F_{\text{BBQ}}(\alpha; \cdots |_{n,t})$. Because $n$ (calibration set size) varies slightly across splits, the *Failed Proportions* value corresponds to the average

$$\left. \frac{1}{|\mathcal{B}|} \sum_{(n,t) \in \mathcal{B}} \left( 1 - F_{\text{BBQ}}\big(\alpha; \cdots |_{n,t}\big) \right) \right|_{\alpha=0.1} \approx 0.46,$$

where $\mathcal{B}$ is the set of different splits, matching the observed 0.44 height. The *Failed Proportions* values of the rest of the Table 1 show across $\delta$, PAC-FCP stays below the target $\delta$ while being noticeably less conservative than CoJER, which reflects its different (uniform-in-$\alpha$) guarantee.

Figure 3 is the visualization of the regression results reported in Table 1. The bars represent the failure rate for regression, i.e., the fraction of splits for which $\frac{K}{t} > \alpha$. Across both $\alpha$ values, PAC-FCP remains at or below the target line $\delta$ while being less conservative than CoJER.

**Prediction interval/set sizes.** Interval sizes are not directly comparable across tasks with different scales. So we normalize each prediction interval length by the length of SCP on the *same* split at the *same* $\alpha$. We do the same for classification tasks as well. The *Average Size* columns of Table 1 report the mean of these ratios; smaller is better. As seen in Table 1, PAC-FCP and CoJER satisfy the empirical guarantee (failure rate $\leq \delta$), but PAC-FCP produces the smallest prediction interval/set size among methods that meet the target FCP coverage.

**Behaviour across $\delta$.** To visualize how failure rate scales vary with $\delta$, Figure 4 plots the empirical failure proportion (left column) and relative prediction interval length (right column) for $\alpha \in \{0.1, 0.2\}$. *Each curve averages over all datasets and models of that type with $n_{split}$ different splits and the shaded band shows the min–max range across four independent repeats of all experiments. PAC-FCP tracks the diagonal $y = \delta$ closely in both regression and classification, indicating neither under- nor over-coverage, while CoJER and Gazin et al. are consistently below the line (conservative) and SCP lies well above (uncontrolled).*

*We note that the discrepancy between smoothness of PAC-FCP's curve in regression and classification curves is primarily due to differences in experimental averaging across runs rather than a fundamental difference in PAC-FCP's behavior. For regression, results are aggregated across multiple datasets (17 OpenML datasets), multiple models, and a relatively small number of splits per dataset, which induces additional variability and leads to looser curves. In contrast, the classification results use a fixed pretrained deep model (ResNet) on a single image classification dataset with a much larger number of splits ($n_{split} = 3000$), resulting in smoother estimates that closely track the target $\delta$ line.*

**The mapping between $\alpha$ and $\alpha'$.** To build intuition about the relation between SCP and the guarantee in Equation (2), we report tables that summarize the mapping observed in our experiments. As noted in Algorithm 1, the computed $\alpha'$ depends on $(n, t)$, which vary across splits in our setup. Table 2 presents the mapping from $\alpha$ to the resulting average $\alpha'$. As expected, $\alpha'$ is smaller than $\alpha$, and their difference reduces as $\delta$ increases.

|  | $\alpha = 0.10$ | $\alpha = 0.20$ |
|---|---|---|
| $\delta = 0.05$ | 0.074 | 0.167 |
| $\delta = 0.10$ | 0.079 | 0.173 |
| $\delta = 0.15$ | 0.084 | 0.178 |
| $\delta = 0.20$ | 0.084 | 0.180 |
| $\delta = 0.25$ | 0.089 | 0.184 |
| $\delta = 0.50$ | 0.099 | 0.199 |

Table 2: The value of $\alpha'$ (in Algorithm 1) averaged over 17 regression tasks $\times$ 5 models $\times$ $n_{\text{split}}$=40.

We present additional results and analyses, along with a more detailed experimental setup, in the Appendix. *Specifically, in Section B.2, we report the exact set sizes in the classification setup; in Section B.3, we investigate the effects of different number of calibration examples on PAC-FCP; and in Section B.4, we provide additional mappings between $\alpha$ and $\alpha'$.*

| α | δ | Regression | | | | | | | |
|---|---|---|---|---|---|---|---|---|---|
| | | SCP | | Gazin et al. | | CoJER | | PAC-FCP | |
| | | Failed pr. | Rel size | Failed pr. | Rel size | Failed pr. | Rel size | Failed pr. | Rel size |
| 0.10 | 0.05 | 0.523 | 1.000 | 0.000 | 2.791 | 0.005 | 1.555 | 0.050 | **1.330** |
| | 0.10 | | | 0.000 | 2.400 | 0.010 | 1.496 | 0.098 | **1.249** |
| | 0.15 | | | 0.000 | 2.189 | 0.019 | 1.426 | 0.147 | **1.197** |
| | 0.20 | | | 0.000 | 2.049 | 0.021 | 1.412 | 0.199 | **1.158** |
| | 0.25 | | | 0.000 | 1.940 | 0.029 | 1.381 | 0.252 | **1.127** |
| | 0.30 | | | 0.000 | 1.854 | 0.039 | 1.357 | 0.297 | **1.100** |
| | 0.35 | | | 0.000 | 1.783 | 0.051 | 1.329 | 0.352 | **1.075** |
| | 0.40 | | | 0.001 | 1.722 | 0.068 | 1.293 | 0.398 | **1.052** |
| | 0.45 | | | 0.002 | 1.669 | 0.074 | 1.279 | 0.454 | **1.030** |
| | 0.50 | | | 0.002 | 1.621 | 0.085 | 1.268 | 0.503 | **1.009** |
| 0.20 | 0.05 | 0.514 | 1.000 | 0.000 | 1.649 | 0.004 | 1.437 | 0.048 | **1.245** |
| | 0.10 | | | 0.001 | 1.551 | 0.007 | 1.386 | 0.101 | **1.186** |
| | 0.15 | | | 0.002 | 1.492 | 0.012 | 1.357 | 0.149 | **1.148** |
| | 0.20 | | | 0.002 | 1.449 | 0.014 | 1.339 | 0.204 | **1.119** |
| | 0.25 | | | 0.005 | 1.414 | 0.017 | 1.324 | 0.251 | **1.096** |
| | 0.30 | | | 0.007 | 1.386 | 0.027 | 1.292 | 0.300 | **1.075** |
| | 0.35 | | | 0.012 | 1.360 | 0.042 | 1.257 | 0.349 | **1.056** |
| | 0.40 | | | 0.015 | 1.338 | 0.052 | 1.244 | 0.396 | **1.039** |
| | 0.45 | | | 0.020 | 1.317 | 0.061 | 1.229 | 0.457 | **1.022** |
| | 0.50 | | | 0.028 | 1.297 | 0.073 | 1.214 | 0.493 | **1.006** |
| α | δ | Classification | | | | | | | |
| | | SCP | | Gazin et al. | | CoJER | | PAC-FCP | |
| | | Failed pr. | Rel size | Failed pr. | Rel size | Failed pr. | Rel size | Failed pr. | Rel size |
| 0.10 | 0.05 | 0.523 | 1.000 | 0.000 | 2.791 | 0.005 | 1.555 | 0.050 | **1.330** |
| | 0.10 | | | 0.000 | 2.400 | 0.010 | 1.496 | 0.098 | **1.249** |
| | 0.15 | | | 0.000 | 2.189 | 0.019 | 1.426 | 0.147 | **1.197** |
| | 0.20 | | | 0.000 | 2.049 | 0.021 | 1.412 | 0.199 | **1.158** |
| | 0.25 | | | 0.000 | 1.940 | 0.029 | 1.381 | 0.252 | **1.127** |
| | 0.30 | | | 0.000 | 1.854 | 0.039 | 1.357 | 0.297 | **1.100** |
| | 0.35 | | | 0.000 | 1.783 | 0.051 | 1.329 | 0.352 | **1.075** |
| | 0.40 | | | 0.001 | 1.722 | 0.068 | 1.293 | 0.398 | **1.052** |
| | 0.45 | | | 0.002 | 1.669 | 0.074 | 1.279 | 0.454 | **1.030** |
| | 0.50 | | | 0.002 | 1.621 | 0.085 | 1.268 | 0.503 | **1.009** |
| 0.20 | 0.05 | 0.514 | 1.000 | 0.000 | 1.649 | 0.004 | 1.437 | 0.048 | **1.245** |
| | 0.10 | | | 0.001 | 1.551 | 0.007 | 1.386 | 0.101 | **1.186** |
| | 0.15 | | | 0.002 | 1.492 | 0.012 | 1.357 | 0.149 | **1.148** |
| | 0.20 | | | 0.002 | 1.449 | 0.014 | 1.339 | 0.204 | **1.119** |
| | 0.25 | | | 0.005 | 1.414 | 0.017 | 1.324 | 0.251 | **1.096** |
| | 0.30 | | | 0.007 | 1.386 | 0.027 | 1.292 | 0.300 | **1.075** |
| | 0.35 | | | 0.012 | 1.360 | 0.042 | 1.257 | 0.349 | **1.056** |
| | 0.40 | | | 0.015 | 1.338 | 0.052 | 1.244 | 0.396 | **1.039** |
| | 0.45 | | | 0.020 | 1.317 | 0.061 | 1.229 | 0.457 | **1.022** |
| | 0.50 | | | 0.028 | 1.297 | 0.073 | 1.214 | 0.493 | **1.006** |

Table 1: *Failure proportions and average prediction interval sizes across all $\alpha$ and $\delta$ values for the four methods: SCP, Gazin et al., CoJER, and PAC-FCP. SCP fails to achieve valid coverage. The uniform-in-$\alpha$ methods are more conservative, but PAC-FCP's failure proportions are much closer to the target $\delta$ value. Moreover, PAC-FCP produces significantly smaller prediction intervals (denoted in bold) compared to CoJER and Gazin et al.*

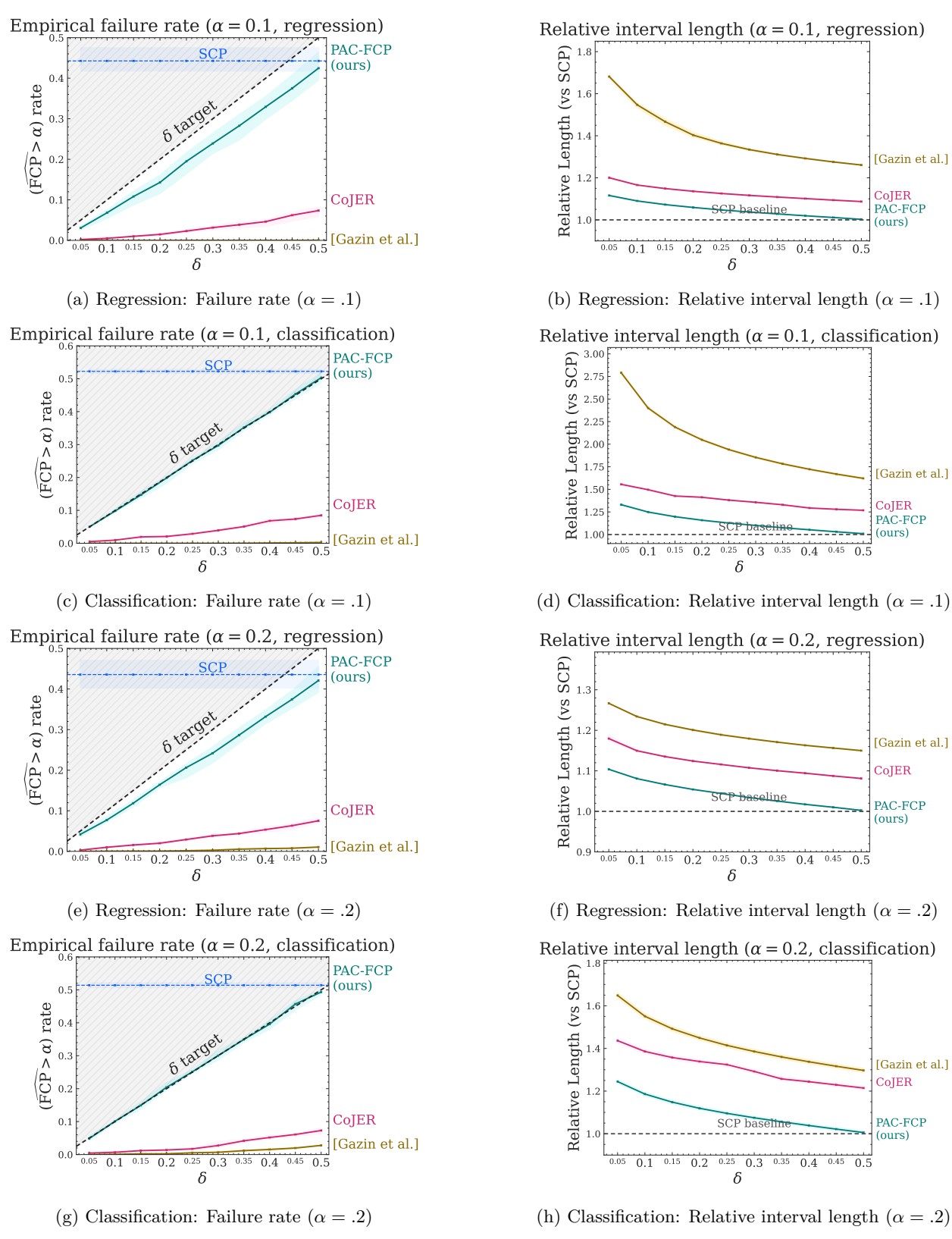

Figure 4: **Comparison across miscoverage levels** $\alpha \in \{0.1, 0.2\}$. Panels (a, b, c, d) use $\alpha = 0.1$ and (e, f, g, h) use $\alpha = 0.2$. The failure rate panels show the proportion of splits among $n_{split}$ randomized experiments with $\widehat{\mathrm{FCP}} > \alpha$ versus $\delta$; the dashed diagonal $y = \delta$ is the target implied by the guarantee in Equation (2). The relative average interval/set size is normalized by SCP at the same $\alpha$ on the same split (baseline = 1.0). The shaded band denotes the range (min–max) across four independent runs for regression and five runs for classification. PAC-FCP closely tracks the $\delta$ target line and yields smaller intervals/sets across all configurations.

### 6.3 Discussion

**Robustness to distribution shift.** *As with standard split CP, PAC-FCP relies on the exchangeability (i.i.d.) assumption between calibration and test data. Our primary goal in this work is to show that, under the same assumptions as vanilla split CP, the batch-level FCP control problem can be reduced to a standard conformal calibration problem via the PAC-FCP method. Studying robustness to distribution shift is out of scope for this paper. Fortunately, PAC-FCP's reduction to standard CP allows us to easily plug-and-play with advanced CP methods to handle specific challenges including distribution shift (e.g., weighted CP methods) as elaborated below.g*

*Importantly, the PAC-FCP approach does not depend on a specific scoring mechanism beyond the typical rank-uniformity property underlying the standard CP. As a result, conformal methods designed to handle distribution shifts to achieve standard CP coverage guarantee could in principle be combined with PAC-FCP, provided they restore the uniform rank assumption for test inputs. In such cases, the resulting PAC-style FCP guarantee would continue to apply. Future work should consider investigating this promising direction both theoretically and empirically.*

## 7 SUMMARY

This paper studies the common but under-explored setting where split conformal prediction (SCP) is calibrated once and then deployed on a batch of $t > 1$ test inputs. SCP guarantees marginal coverage for a single testing input but says nothing about the realized false coverage proportion (FCP) across that batch. We present a simple and effective method, PAC-FCP, that provides a high probability guarantee for FCP by leveraging vanilla SCP methods. This retains SCP's simplicity, while giving practitioners exactly the $(\alpha, \delta)$-FCP control they often need. Experiments on multiple benchmark regression and classification tasks demonstrate that PAC-FCP hits the desired failure rate ($\approx \delta$) and yields significantly smaller prediction intervals/sets than existing methods.

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

## A  PROOFS

**Proof of Theorem 1.**  Here we provide the complete proof of Theorem 1, following *Vovk (2012); Angelopoulos & Bates (2022); Gazin et al. (2024); Marques F. (2025)* as this result is not commonly used in CP research. The intuition behind the proof is that, in the batch setting and *conditional* on $\mathcal{D}_{\text{cal}}$, the test outcomes are Bernoulli trials, and the calibration conditional coverage follows a Beta distribution because of the rank statistics.

**Theorem 1** (Batch miscoverage is Beta–Binomial)**.** *Let $C_\alpha$ be a conformal method with level $\alpha$, on i.i.d data, with a continuous conformity score (defined as $S$) and randomized ties. Let $Z_j = \mathbf{1}\{Y_{n+j} \notin C_\alpha(X_{n+j})\}$ for $j = 1, \ldots, t$, and $K = \sum_{j=1}^t Z_j$. And define $\ell(\alpha) := \lfloor \alpha(n+1) \rfloor$.*

*The predictive distribution of $K$ over an evaluation batch of size $t$ is Beta-Binomial.*

$$\Pr(K = k) \;=\; \binom{t}{k} \frac{B(k + \ell, \; t - k + n + 1 - \ell)}{B(\ell, \; n + 1 - \ell)},$$

*where $B(\cdot, \cdot)$ is the Beta function and $k \in \{0, \ldots, t\}$.*

*Proof.* For $S$ with continuous CDF of $F$, define' random variable $U$ as $U := F(S(X, Y))$, because it has no atoms, has a uniform distribution. Since $F$ is increasing, the conformal set is

$$y \in C(x) \iff S(x, y) \leq q_{\text{conformal}} \iff F(S(x, y)) \leq U_{(n-l)} \iff U \leq U_{(n-l)} \tag{4}$$

*where* $U_{(k)}$ is the $k$-th order statistics of $U_i$s.

Fix $\alpha$ and a calibration set $D_{\text{cal}} = \mathcal{D}_{\text{cal},0}$. Define the conformal set *as*

$$C_\alpha(x) \;:=\; \{\, y \in \mathcal{Y} : \; S(x, y) \leq q(\mathcal{D}_{\text{cal},0}, \alpha) \,\},$$

where $q(\mathcal{D}_{\text{cal},0}, \alpha)$ is the usual $n+1$ order–statistic quantile computed from the calibration scores $\{S_i\}_{i=1}^n$. Thus, $C_\alpha(\cdot)$ is a deterministic function of $x$ given $\mathcal{D}_{\text{cal},0}$, and so is

$$W(x, y) \;:=\; \mathbf{1}\{\, y \notin C_\alpha(x) \,\}.$$

Now sample $(X_{n+j}, Y_{n+j}) \overset{\text{i.i.d.}}{\sim} \mathcal{X} \times \mathcal{Y}$ for $j = 1, \ldots, t$. Conditional on $D_{\text{cal}} = \mathcal{D}_{\text{cal},0}$, the variables

$$Z_j \;:=\; W(X_{n+j}, Y_{n+j}) \;=\; \mathbf{1}\{\, Y_{n+j} \notin C_\alpha(X_{n+j}) \,\}$$

are i.i.d. Bernoulli with success probability $1 - \theta$. *$\theta$ defined as shown below*

$$\theta \;=\; \Pr\big(Y \in C_\alpha(X) \,\big|\, \mathcal{D}_{\text{cal}} = \mathcal{D}_{\text{cal},0}\big)$$

is the calibration-conditional coverage. Using equation 4,

$$\theta = \Pr\left(U < U_{(n-l)} | \{U_i\}_{i=1}^n\right)$$

is the CDF of $n - l$ order statistics of $n$ uniform variable, which is a Beta distribution with $n + 1 - l$ and $l$ parameters. Conditioned on $\mathcal{D}_{\text{cal}} = \mathcal{D}_{\text{cal},0}$,

$$K \;=\; \sum_{j=1}^t Z_j \;\sim\; \text{Binomial}\big(t, \, 1 - \theta\big).$$

By the law of total probability,

$$\Pr\big(K = k\big) \;=\; \mathbb{E}\big[\Pr\big(K = k \,\big|\, \mathcal{D}_{\text{cal}}\big)\big] \;=\; \mathbb{E}\big[\Pr\big(K = k \,\big|\, \theta\big)\big].$$

Taking the outer expectation with respect to $\theta$—which satisfies $\theta \sim \text{Beta}(n + 1 - \ell, \ell)$ with $\ell = \lfloor \alpha(n+1) \rfloor$—amounts to integrating the Binomial pmf against the Beta pdf. This yields the Beta–Binomial pmf, i.e.,

$$K \;\sim\; \text{Beta-Binomial}\big(t, \, \ell, \, n+1 - \ell\big),$$

which is the claim. □

**Proof of Lemma 2.**

**Lemma 2** (PAC-style FCP control via tail inversion)**.** *Fix $\alpha \in [0,1]$, $\delta \in (0,1)$, and batch size of testing inputs $t$. Let $\ell(\alpha') := \lfloor \alpha'(n+1) \rfloor$ for some internal split CP level $\alpha'$. If*

$$\sum_{k=0}^{\lfloor \alpha t \rfloor} \binom{t}{k} \frac{B(k+\ell,\ t-k+n+1-\ell)}{B(\ell,\ n+1-\ell)} \ \geq \ 1 - \delta, \tag{$\star$}$$

*Then, split CP run at level $\alpha'$ with a score that does not produce ties (e.g., via random tie-breaking) guarantees*

$$\Pr\left(\frac{K}{t} \leq \alpha\right) \ \geq \ 1 - \delta.$$

*Proof.* We start with Equation ($\star$) in Lemma 2:

$$\sum_{k=0}^{\lfloor \alpha t \rfloor} \binom{t}{k} \frac{B(k+\ell,\ t-k+n+1-\ell)}{B(\ell,\ n+1-\ell)} \ \geq \ 1 - \delta, \tag{$\star$}$$

where $\ell = \lfloor \alpha'(n+1) \rfloor$.

Substituting the Beta–Binomial pmf from Theorem 1, the left-hand side becomes

$$\sum_{k=0}^{\lfloor \alpha t \rfloor} \Pr(K = k),$$

which is precisely the CDF of the Beta-Binomial$(t, \ell, n+1-\ell)$ distribution evaluated at $k = \lfloor \alpha t \rfloor$. Hence Equation ($\star$) is equivalent to

$$\Pr(K/t \leq \alpha) \ \geq \ 1 - \delta,$$

which establishes the claim. $\qquad \square$

**Corollary 3.** *For a method that generates sets using a noncoformity score, and compare that to a threshold calculated from a quantile of the scores $S_1, \cdots S_n$ called q, that gets the guarantee in Eq. 2, we have:*

$$q \leq q^*$$

*Where $q^*$ is the $\alpha^*$ conformal quantile (i.e. n+1 corrected) of the calibration scores, where $\alpha^*$ is the maximum $\alpha'$ satisfying the condition equation $\star$ in Lemma 2.*

*Proof.* For standard monotone non-conformity scores (e.g., absolute residuals in regression), the set

$$C_{\alpha'}(x) = \{\, y \in \mathcal{Y} : s(x,y) \leq q_{1-\alpha'} \,\}$$

shrinks as $\alpha'$ increases, so the conformal threshold $q_{1-\alpha'}$ - the $(1 - \alpha')$ quantile computed with the $n+1$ order-statistic rule - is strictly decreasing in $\alpha'$.

Among all SCP-like procedures that satisfy Equation ($\star$), choosing the *largest* feasible $\alpha'$ therefore yields the smallest $q_{1-\alpha'}$ and, hence, the narrowest prediction sets. $\qquad \square$

## B  EXPERIMENTAL RESULTS

### B.1  Experimental Setup.

All experiments were run on 32 CPU threads (AMD EPYC 7573X 32-Core) with 250 GiB RAM. And the classification inference was done on a single NVIDIA T4 GPU. Our code forks and uses the official CoJER implementation. We implement the BBQ function $F_{\text{BBQ}}$ with `scipy` and, for Step 5 of Algorithm 1, perform a grid search over the maximum $\alpha'$ values at each discrete step (cf. the left plot of Figure 2). For each configuration we collect results in groups of $n_{\text{split}} = 10$ with different random seeds (listed in the code), and aggregate the runs to produce the tables and figures in this paper. For classification, we use $n_{\text{split}} = 3000$, and we use the APS score.

## B.2   Exact set sizes

We provide results for PAC-FCP and compare them to the two baselines mentioned in Section 6. The experiments are run for $\alpha \in \{0.1, 0.2\}$ and $\delta \in \{0.05, 0.10, 0.15, 0.20, 0.25, 0.30, 0.35, 0.40, 0.45, 0.50\}$.

The reported sizes in the main paper are relative to SCP. For our regression task, this makes sense because there are multiple data distributions. For classification, since we only have one model and one distribution, we report the actual sizes in Table 3, which is the analogue of Table 1 in the main paper, but only for classification and with the actual set sizes.

| $\alpha$ | $\delta$ | SCP | | CoJER | | PAC-FCP | |
|---|---|---|---|---|---|---|---|
| | | Failed prop. | Avg size | Failed prop. | Avg size | Failed prop. | Avg size |
| 0.10 | 0.05 | 0.523 | 12.757 | 0.005 | 19.843 | 0.050 | 16.968 |
| | 0.10 | 0.523 | 12.757 | 0.010 | 19.084 | 0.098 | 15.931 |
| | 0.15 | 0.523 | 12.757 | 0.019 | 18.194 | 0.147 | 15.273 |
| | 0.20 | 0.523 | 12.757 | 0.021 | 18.016 | 0.199 | 14.777 |
| | 0.25 | 0.523 | 12.757 | 0.029 | 17.615 | 0.252 | 14.374 |
| | 0.30 | 0.523 | 12.757 | 0.039 | 17.307 | 0.297 | 14.036 |
| | 0.35 | 0.523 | 12.757 | 0.051 | 16.956 | 0.352 | 13.714 |
| | 0.40 | 0.523 | 12.757 | 0.068 | 16.501 | 0.398 | 13.423 |
| | 0.45 | 0.523 | 12.757 | 0.074 | 16.319 | 0.454 | 13.141 |
| | 0.50 | 0.523 | 12.757 | 0.085 | 16.176 | 0.503 | 12.872 |
| 0.20 | 0.05 | 0.514 | 5.187 | 0.004 | 7.451 | 0.048 | 6.455 |
| | 0.10 | 0.514 | 5.187 | 0.007 | 7.188 | 0.101 | 6.152 |
| | 0.15 | 0.514 | 5.187 | 0.012 | 7.040 | 0.149 | 5.956 |
| | 0.20 | 0.514 | 5.187 | 0.014 | 6.944 | 0.204 | 5.805 |
| | 0.25 | 0.514 | 5.187 | 0.017 | 6.869 | 0.251 | 5.683 |
| | 0.30 | 0.514 | 5.187 | 0.027 | 6.702 | 0.300 | 5.575 |
| | 0.35 | 0.514 | 5.187 | 0.042 | 6.521 | 0.349 | 5.478 |
| | 0.40 | 0.514 | 5.187 | 0.052 | 6.453 | 0.396 | 5.387 |
| | 0.45 | 0.514 | 5.187 | 0.061 | 6.376 | 0.457 | 5.300 |
| | 0.50 | 0.514 | 5.187 | 0.073 | 6.299 | 0.493 | 5.216 |

Table 3: Failure proportions and average set sizes for classification methods

## B.3   Sensitivity to number of calibration examples.

*To demonstrate the sensitivity of PAC-FCP to different calibration set sizes, we present the results of a simple experiment. We use the same model and data as in our original classification experiments, and for different values of the calibration size n, we report the failed proportion (similar to Table 1) as well as the actual average prediction set size. We use $\alpha = 0.1$ and $\delta = 0.1$, with a batch size of $t = 500$. The results of this experiment are reported for 500 independent splits in Table 4.*

| $n$ | Failed prop. | Avg size |
|---|---|---|
| 100 | 0.072 | 26.28 |
| 500 | 0.076 | 17.98 |
| 1000 | 0.068 | 16.68 |
| 5000 | 0.108 | 16.19 |
| 10000 | 0.090 | 16.15 |
| 15000 | 0.114 | 15.96 |
| 20000 | 0.076 | 15.79 |

Table 4: Failure proportions and average set sizes for different calibration sizes. The set sizes generally decrease with additional calibration data. The failed proportion column reports the proportion of experiments that failed the empirical FCP control at level $\alpha$. As observed, this value is maintained around $\delta = 0.1$.

| $\alpha$ | $\delta$ | $n = 100$ | | | $n = 1000$ | | | $n = 10000$ | | | $n = 100000$ | | |
|---|---|---|---|---|---|---|---|---|---|---|---|---|---|
| | | $t=100$ | $t=500$ | $t=1000$ | $t=100$ | $t=500$ | $t=1000$ | $t=100$ | $t=500$ | $t=1000$ | $t=100$ | $t=500$ | $t=1000$ |
| | 0.1 | 0.059 | 0.069 | 0.069 | 0.061 | 0.079 | 0.083 | 0.063 | 0.083 | 0.087 | 0.063 | 0.083 | 0.088 |
| 0.1 | 0.05 | 0.049 | 0.059 | 0.059 | 0.053 | 0.074 | 0.079 | 0.055 | 0.078 | 0.084 | 0.055 | 0.079 | 0.085 |
| | 0.01 | 0.029 | 0.039 | 0.039 | 0.040 | 0.065 | 0.071 | 0.042 | 0.070 | 0.078 | 0.042 | 0.071 | 0.079 |
| | 0.1 | 0.019 | 0.029 | 0.029 | 0.023 | 0.035 | 0.038 | 0.024 | 0.038 | 0.041 | 0.025 | 0.038 | 0.041 |
| 0.05 | 0.05 | 0.019 | 0.019 | 0.029 | 0.018 | 0.032 | 0.035 | 0.020 | 0.035 | 0.039 | 0.020 | 0.035 | 0.039 |
| | 0.01 | – | 0.019 | 0.019 | 0.011 | 0.026 | 0.029 | 0.013 | 0.030 | 0.035 | 0.013 | 0.030 | 0.035 |

Table 5: $\alpha'$ values based on values of $n$, $t$, $\alpha$ and $\delta$. Only in the extreme case $\delta = 0.01$ and $\alpha = 0.05$, with the smallest batch and calibration sizes, PAC-FCP could not find a positive $\alpha'$.

### B.4   Additional mapping between $\alpha$ and $\alpha'$.

*In the main paper, in Table 2, we provided empirically averaged mappings between $\alpha$ and $\alpha'$ across our evaluation setup (i.e., different calibration sizes and different evaluation batch sizes). Here, to provide more intuition, we provide Table 5, which reports the values of $\alpha'$ for different choices of $\alpha$, $\delta$, calibration size $n$, and evaluation batch size $t$.*

