# OpenReview forum: "Controlling Coverage of Uncertainty Sets for Batch Evaluation via Vanilla Conformal Prediction"
_TMLR — Rejected by TMLR_

### Review · Reviewer_cLxp · 2025-12-08

**Summary Of Contributions:**

This paper proposes a new conformal prediction-based approach, PAC-FCP, for controlling the error rates during batch evaluation. This approach is based on the observation that coverage follows a beta-binomial distribution, allowing substantially tighter intervals than CoJER, the most recent method which PAC-FCP is benchmarked against.

Coming from the perspective of a practitioner, I find the framing and approach compelling. The limitations of standard SCP definitely do come up when working with downstream (non stats/ML) researchers, and I can see this being reasonably used in practice. I also agree that the cost of such an approach (fixing $\alpha$, $\delta$, $t$) is very reasonable in practice, and would come with few objections. Overall, I would personally like to see this paper accepted so that this idea can be put to test by the wider community.

Strengths:
- The paper is well-written and easy to read compared to other papers with similar content and depth
- The proposed procedure provides smaller prediction intervals (on the provided benchmarks) than state-of-the art baselines …
- … while being much simpler, and overall seems like it could be reasonable for even a non-machine-learning researcher to implement with some guidance.

Weaknesses:
- The $\alpha$ and $\delta$ values tested are fairly large, at least in my experience, though this seems to be the norm given prior papers concerning this problem.
- The algorithm relies on a grid search over $\alpha’$. This seems like it shouldn’t be a big deal (both in computational and software complexity), though it’s unclear to me whether this is indeed the case, especially if the various parameters / sizes were to be changed by several orders of magnitude (e.g., very small $\alpha$ / very large $n$)

**Audience:**

Yes

**Audience Explanation:**

Yes; I believe researchers developing and using conformal prediction techniques would be interested in the findings of this paper. The key insight seems simple, but novel, and the paper gives attention to an important subfield which has not gotten much attention in the past.

**Broader Impact Concerns:**

This paper concerns fundamental statistical machine learning methodology without any ethical concerns or direct societal impact.

**Claims And Evidence:**

Yes

**Claims Explanation:**

The claims in this paper are supported by mathematical proofs and empirical evaluations which mostly follow the procedure used by CoJER (Blain et al., 2025). The empirical evaluations are fairly complete, and cover all the key points I can think of (though the paper can of course always benefit from more/larger evaluations). I haven’t been able to spot any clear errors or omissions in the proofs, though I am not really an expert in this area, and it’s very possible that I’ve missed something.

**Requested Changes:**

Coming from the perspective of a practitioner in this area, not a researcher, here are a few things I believe would strengthen the paper:
- It would be helpful to see code samples. In addition to a useful artifact for practitioners, this could help to get a more concrete view of the relative implementation complexity of this method, since mathematical descriptions can sometimes appear deceptively simple. I suspect this will actually make the relative simplicity of PAC-FCP even more apparent.
- It would be good to see some evaluations which push PAC-FCP to its limits, like extreme values of $\alpha$, $\delta$, and so-on. For example, what would happen on a (possibly simulated) regression task with $\alpha < 10^{-3}$ but a calibration set with $n > 10^5$ (or whatever value is reasonable)?

More broadly, I don’t think this paper requires any changes (unless another reviewer finds a substantive error), and I’m happy with it as is.

---

> ### Author Response · Authors · 2026-01-04
> **Response to Reviewer cLxp**
>
> We thank the reviewer for their thoughtful and encouraging feedback, and for evaluating the paper from a practitioner’s perspective. We are glad that the framing and practical motivation resonate, and we appreciate the constructive suggestions.
>
> ## Code availability and implementation clarity.
>
> We agree that concrete code examples can significantly improve accessibility and help practitioners better assess implementation complexity. Therefore, we have added the full experimental code used in this paper to the supplementary materials and will release the code on a public GitHub repository at the time of publication. We believe this makes the simplicity of the proposed procedure more transparent and will facilitate adoption by practitioners.
>
> ## Behavior under extreme parameter regimes.
>
> We thank the reviewer for this insightful question. Studying prediction set sizes under extreme values of α, δ, n, and t using fully simulated distributions can be challenging, as such simulations may not reflect the behavior of conformal prediction on realistic data distributions. That said, the dependence of the internal level α′ on α, n, and t follows directly from the theoretical characterization underlying the method.
>
> To provide additional intuition, we added a new Appendix B.4 in the revised paper that empirically studies the effect of n, t, and α on prediction set sizes across a range of parameter settings, including more extreme regimes (e.g., very small α and large n).
>
> We thank the reviewer again for their positive assessment and helpful suggestions, which we believe strengthened the paper.

---

### Review · Reviewer_trjT · 2025-12-10

**Summary Of Contributions:**

The paper proposes a method to obtain uncertainty sets for multiple test inputs with a fixed calibration set. The presented method enjoys theoretical guarantees that ensure that the False Coverage Proportion (FCP) is controlled with high probability. Experimental results are provided, showing that the proposed method achieves the specified FCP rate with small prediction sets/intervals.

Strengths:
- The experiments show that the proposed method generates smaller prediction sets than previous methods.

Weaknesses:
- The authors claim that the presented method is novel, however it seems to be already known in the literature (see discussion below).
- The quality of the writing is not as high as it should be.

**Audience:**

No

**Audience Explanation:**

The presented methodology seems to be known, and the quality and clarity of writing are not as high as they should be.

Theorem 1 in the presented paper seems to be already known (Corollary A.3 in [1], Theorem 1 in [2] and Theorem 3 in [3]). Lemma 2 in the presented paper seems to be a particular case of Theorem B.1 in [1]. Moreover, the proposed method/algorithm seems to coincide with the "beta template" choice in Theorem B.1 in [1].

[1] U. Gazin, G. Blanchard, and E. Roquain. "Transductive Conformal Inference with Adaptive Scores." AISTATS, 2024.

[2] P. C. Marques F. (2023). "On the Universal Distribution of the Coverage in Split Conformal Prediction." arXiv:2303.02770, 2023.

[3] K. Huang, Y. Jin, E. Candes, and J. Leskovec. "Uncertainty Quantification Over Graphs with Conformalized Graph Neural Networks." NeurIPS, 2023.

The current text has many typos. For instance:
- The subscript $\alpha$ is missing in $C(X_{n+1})$ in the last equation in page 3.
- $\{ (X_i,Y_i)\}_{i=1}^n \sim (\mathcal{X},\mathcal{Y})^n$ in page 4.
- Periods are missing in the proof of Theorem 1.
- “We employ$n_{\text{split}}$” in page 8.
- The notations Pr and $\mathbb{P}$ are used arbitrarily all over the paper.

There are many serious issues with the references. They are not written in a consistent manner and even there are duplicated references: (Gazin et al. (2024a) and Gazin et al. (2024b)). Also, there are typos in the references:

> Paulo C. Marques F. Universal distribution of the empirical coverage in split conformal prediction. Statistics & Probability Letters, 219(C):**None, None** 2025.  …

**Claims And Evidence:**

Yes

**Claims Explanation:**

Proofs are presented in the Appendix and experimental results are explained in detail.

**Requested Changes:**

- Compare with [1]. At the present point, it seems that all the results and algorithm from the paper are present in [1].

- The quality of the writing should be improved: fix typos and issues with the references.

- The clarity of the paper should be improved. For instance, the notations $\mathcal{C}(\alpha^*)$ and $C_k(x)$ in page 4 are not introduced.

All the proposed adjustments are critical to securing recommendation for acceptance.

[1] U. Gazin, G. Blanchard, and E. Roquain. "Transductive Conformal Inference with Adaptive Scores." AISTATS, 2024.

---

> ### Author Response · Authors · 2026-01-04
> **Response to Reviewer trjT**
>
> We thank the reviewer for their careful reading of the paper and for raising important points regarding prior work, clarity, and presentation quality. We address these issues below.
>
> ## Novelty and relationship to prior work.
>
> **Part 1:** We agree that the key theoretical result used in this paper (Theorem 1) is known and has been previously analyzed. In fact, in our original submission, we cited prior work [R1] [R2] that has Theorem 1 and explicitly acknowledged this fact in one of our contributions bullet: “Theoretical analysis of PAC-FCP for validity and effectiveness to control FCP by building on prior theoretical results.”
>
> The paper’s contribution is to effectively use this known theoretical result to construct a simple, practitioner-oriented CP framework for batch evaluation. Specifically, the proposed PAC-FCP method shows how Theorem 1 can be operationalized to produce an explicit PAC-style guarantee on the realized false coverage proportion for a fixed evaluation batch, while retaining the simplicity of vanilla split conformal prediction.
>
> In the revised paper, we have cited [R1] [R2] [R3] [R4] to acknowledge that Theorem 1 is a known result and that the PAC-FCP algorithm leverages it to solve a practical problem setting.
>
> **Part 2:** We would like to clarify that Lemma 2 in our paper is a direct consequence of Theorem 1 (Batch Miscoverage is Beta Binomial) from prior work.
>
> Thanks for pointing us to Theorem B.1 from the Appendix of Gazin et al., 2024 paper [R3]. First, this paper is primarily geared towards a theoretical investigation of the joint distribution of conformal p-values and provides a concrete approach for uniform-in-α guarantee for false coverage proportion (FCP) over a batch of test inputs. Second, the “beta template” mentioned in the appendix is not investigated theoretically and empirically with the main algorithmic approach. To the best of our knowledge, there is no practical implementation of the variant with “beta template” (https://github.com/ulyssegazin/TransductiveAdaptive_CP) to perform experiments.
>
> We spent quite a bit of time reading the two papers (Gazin et al., 2024 paper [R3] and Blanchard et al., 2023 [R5]). However, we could not identify a direct connection between our PAC-FCP method and the beta template version of the approach from Gazin et al., 2024 [R3]. If the reviewer knows the direct connection precisely and can articulate it, we will be happy to incorporate this connection in the revised paper. In any case, our paper empirically demonstrates the effectiveness of PAC-FCP in producing smaller uncertainty sets over prior methods with uniform-in-α guarantee (i.e., Gazin et al., 2024 [R3] and Blain et al., 2025 [R6]).
>
> In the revised paper, we have added results for the approach in Gazin et al., 2024 [R3] based on their code and the CoJER library (Blain et al., 2025 [R6]). As mentioned in Blain et al., 2025 [R6], the implementation of Gazin et al., 2024 [R3] provides a uniform-in-alpha guarantee (similar to CoJER) but is more conservative.
>
> ## Writing quality, notation, and references.
>
> We thank the reviewer for pointing out the typos, notational inconsistencies, and reference issues. We agree that these issues affect clarity. In the revised paper, we have corrected the reported typos, standardized probabilistic notation throughout the paper, and fixed inconsistencies and duplication in the references.
>
> We appreciate the reviewer’s detailed feedback and insightful comments based on deep knowledge of the relevant literature. This feedback helped us greatly improve the clarity and presentation of the paper.
>
> [R1] Vladimir Vovk. Conditional validity of inductive conformal predictors, 2012.
>
> [R2] Anastasios N. Angelopoulos, Stephen Bates. A Gentle Introduction to Conformal Prediction and Distribution-Free Uncertainty Quantification, https://arxiv.org/abs/2107.07511, 2021
>
> [R3] U. Gazin, G. Blanchard, and E. Roquain. Transductive Conformal Inference with Adaptive Scores. AISTATS, 2024.
>
> [R4] Paulo C. Marques F. Universal distribution of the empirical coverage in split conformal prediction. Statistics & Probability Letters, 219:110350, 2025.
>
> [R5] Blanchard, G., Neuvial, P., and Roquain, E.. Post hoc confidence bounds on false positives
> using reference families. Annals of Statistics, 48(3):1281–1303, 2020.
>
> [R6] A. Blain, B. Thirion, P. Neuvial. False Coverage Proportion Control for Conformal Prediction. ICML, 2025.

---

> > ### Comment · Reviewer_trjT · 2026-01-09
> >
> > Thank you for the authors’ responses. I still have concerns regarding the novelty of both the results and the proposed method.
> >
> > Specifically, Theorem 1 is already known, and Lemma 2 appears to be a particular case of Theorem B.1 in [1]. Note that: $$F_{\text{BBQ}}(\alpha') = \mathbb{P}(\widehat{F}_m(\alpha')\leq \alpha)$$ (the r.h.s. uses notation from [1]). Moreover, $\lambda(\delta,n,m)$ coincides with steps 5-9 in Algorithm 1. Hence, the method presented in the paper is a particular case of the beta template from [1] with $\mathcal{K} = \alpha t$. Consequently, some of the discussion in the revised paper appears incorrect and should be adjusted.
> >
> > At present, the manuscript does not clearly discuss the novelty of the results or method. I recommend explicitly citing prior work (including paper and theorem numbers) and clarifying how the present contributions extend or differ from these existing results.
> >
> > [1] U. Gazin, G. Blanchard, and E. Roquain. "Transductive Conformal Inference with Adaptive Scores." AISTATS, 2024.

---

> > > ### Author Response · Authors · 2026-01-16
> > >
> > > We thank the reviewer for their time, their careful response, and for pointing out the precise theoretical connection.
> > >
> > > We have updated the manuscript to clarify the relationship between Lemma 2 and Theorem B.1 in [1] and have explicitly cited the relevant prior theorems.
> > >
> > > [1] U. Gazin, G. Blanchard, and E. Roquain. Transductive Conformal Inference with Adaptive Scores. AISTATS, 2024.

---

> > > > ### Comment · Reviewer_trjT · 2026-01-19
> > > >
> > > > While the authors have updated the manuscript, the analysis remains insufficiently detailed. For instance, the use of vague phrasing such as “appears to be” weakens the rigor of the conclusions and should be replaced with precise, evidence-backed statements (see my comments in the previous answer).

---

### Review · Reviewer_srJo · 2025-12-20

**Summary Of Contributions:**

This paper proposes PAC-FCP, a method that gives a batch-level false coverage proportion guarantee in conformal prediction. The key idea is that batch miscoverage counts follow a Beta-Binomial distribution, enabling a PAC-style $(\alpha, \delta)$ control without altering standard conformal machinery. Experiments show that this method has certain effectiveness. The authors are recommended to address the following comments to improve the significance of the study and for the benefit of a wider audience.

1. How robust is PAC-FCP to mild distribution shift in the test data? Since the guarantee relies on i.i.d. assumptions and calibration-conditional Beta-Binomial structure, can the authors clarify what happens when the test distribution deviates slightly from the calibration distribution?

2. In Fig. 4, the classification curves closely track the target $\delta$ line, but the regression curves appear noticeably looser. Can the authors explain why PAC-FCP behaves differently across these two settings, and what drives this discrepancy?

3. Given that $\alpha′$ and $l(\alpha′)$ depend directly on the calibration set size $n$, how sensitive is PAC-FCP to $n$?

**Audience:**

Yes

**Audience Explanation:**

The findings of this paper would be of clear interest to a substantial portion of TMLR's audience, particularly researchers working on conformal prediction, uncertainty quantification, and reliable machine learning.

**Broader Impact Concerns:**

The paper is purely methodological and aims to improve the reliability of uncertainty quantification, with no foreseeable negative ethical or societal impacts beyond standard assumptions common to conformal prediction methods.

**Claims And Evidence:**

Yes

**Claims Explanation:**

The main claims of the paper are supported by clear, accurate, and convincing evidence. The theoretical guarantee of PAC-style FCP control is rigorously derived by leveraging the Beta-Binomial characterization of batch miscoverage under split conformal prediction, with all assumptions stated explicitly and proofs provided in detail. The experimental evaluation is well aligned with the stated claims, directly measuring batch-level FCP rather than marginal coverage, and is conducted across a diverse set of regression and classification benchmarks. Overall, the theoretical analysis and empirical results jointly provide strong and coherent support for the paper’s claims.

**Requested Changes:**

Please address the issues raised in the Summary of Contributions section.

---

> ### Author Response · Authors · 2026-01-04
> **Response to Reviewer srJo**
>
> We thank the reviewer for their careful reading of the paper, the positive assessment of our contributions, and the constructive questions. We address each point below.
>
> ## Robustness to distribution shift.
>
> As with standard split conformal prediction, PAC-FCP relies on the exchangeability (i.i.d.) assumption between calibration and test data. Our primary goal in this work is to show that, under the same assumptions as vanilla split CP, the batch-level FCP control problem can be reduced to a standard conformal calibration problem via the PAC-FCP method. Studying robustness to distribution shift is out of scope for this paper. However, due to the reduction to standard CP, we can easily plug-and-play with advanced CP methods to handle specific challenges, including distribution shift (e.g., weighted CP methods) as elaborated below.
>
> Importantly, the PAC-FCP approach does not depend on a specific scoring mechanism beyond the typical rank-uniformity property underlying the standard CP. As a result, conformal methods designed to handle distribution shifts to achieve standard CP coverage guarantee could, in principle, be combined with PAC-FCP, provided they restore the uniform rank assumption for test inputs. In such cases, the resulting PAC-style FCP guarantee would continue to apply.
>
> We have added this clarification and discussion in Section 6.3 of the revised paper.
>
> ## Difference between regression and classification behavior in Figure 4.
>
> The apparent discrepancy between regression and classification curves in Figure 4 is primarily due to differences in experimental averaging across runs rather than a fundamental difference in PAC-FCP behavior.
>
> For regression, results are gathered across multiple datasets (17 OpenML datasets), multiple models, and a relatively small number of splits per dataset (10 splits for Figure 4), which induces additional variability and leads to looser curves. In contrast, the classification results use a fixed pretrained deep model (ResNet) on a single image classification dataset with a much larger number of splits ($n_\text{split} = 3000$), resulting in smoother estimates that closely track the target δ line.
>
> We have added an explicit clarification of this point in Section 6.2 of the revised paper.
>
> ## Sensitivity to calibration set size n.
>
> As with other CP-based methods, it is difficult to characterize sensitivity to n in a distribution-agnostic manner, since key properties of the procedure (e.g., prediction set sizes) depend strongly on the underlying data distribution.
>
> To provide intuition, the original paper included an empirical mapping between α and the internal level α′ (Table 2), which reflects the dependence on n, t (averaged over different values of n and t) implied by the main theorem. To address the reviewer’s question, we have now included a new table (Table 4 in Appendix B.3) in the revised manuscript, which shows the values of set sizes for different choices of α, n, and t.

---

### Decision · Action_Editor_Jo8S · 2026-01-26

**Recommendation:** Reject

**Audience:**

Yes

**Audience Explanation:**

Conformal Prediction and related methods are of great interest to the TMLR community.

**Claims And Evidence:**

No

**Claims Explanation:**

The paper is well-written, but does not clearly and rigorously establish the relationship between its main results and the existing literature, making it hard to verify what is being contributed and which claims are supported. As a result, I’m not yet convinced the paper, in its present form, meets TMLR’s criteria.

The authors will be able to resubmit provided that they (1) include a result-by-result correspondence showing which results are direct corollaries, which require new arguments, and exactly what assumptions differ, (2) clearly state what a TMLR reader gains that they wouldn’t already get from the cited prior methods, or (3) frame the work as a review paper with some new extensions of some operational interest.

**Resubmission Of Major Revision:**

The authors may consider submitting a major revision at a later time.